# Preclinical PET and SPECT Imaging in Small Animals: Technologies, Challenges and Translational Impact

**DOI:** 10.3390/cells15010073

**Published:** 2025-12-31

**Authors:** Magdalena Bruzgo-Grzybko, Izabela Suwda Kalita, Adam Jan Olichwier, Natalia Bielicka, Ewa Chabielska, Anna Gromotowicz-Poplawska

**Affiliations:** 1Department of Biopharmacy and Radiopharmacy, Medical University of Bialystok, 15-222 Bialystok, Poland; natalia.bielicka@umb.edu.pl (N.B.); ewa.chabielska@umb.edu.pl (E.C.); anna.gromotowicz-poplawska@umb.edu.pl (A.G.-P.); 2Radiopharmacy Centre, Medical University of Bialystok, 15-569 Bialystok, Poland; izabela.kalita@umb.edu.pl (I.S.K.); adam.olichwier@umb.edu.pl (A.J.O.)

**Keywords:** small-animal imaging, PET, SPECT, preclinical research, radionuclides, radiopharmaceuticals, molecular imaging, quality control, translational medicine

## Abstract

**Highlights:**

**What are the main findings?**
Advances in PET and SPECT technology significantly improved spatial resolution, sensitivity and quantitative accuracy in small-animal imaging.Experimental conditions and QC procedures critically affect data reliability and reproducibility in preclinical molecular imaging.

**What are the implications of the main findings?**
Enhanced imaging performance increases the translational value of animal models by enabling more precise assessment of biological processes relevant to human disease.Technological progress in multimodal and hybrid systems expands the scope of preclinical research, allowing comprehensive functional–anatomical assessment within a single imaging workflow.

**Abstract:**

Molecular imaging in preclinical research using PET and SPECT has become a key component of contemporary biomedicine, enabling noninvasive, quantitative, and longitudinal assessment of biological processes in vivo. Rapid technological progress, including advances in detector design, readout electronics, reconstruction algorithms, and multimodal integration, has substantially improved spatial resolution, sensitivity, and quantitative accuracy, thereby enhancing the translational value of animal models. PET and SPECT enable precise characterization of metabolic, molecular, and functional alterations across a wide range of diseases including cancer, cardiovascular disorders, neurodegeneration, and inflammation. Radiopharmaceuticals targeting diverse biological pathways, combined with PET and SPECT systems, allow comprehensive and physiologically relevant evaluation of disease mechanisms and therapeutic responses. Despite these significant advances, important challenges remain, including limitations in quantitative precision, partial-volume effects and inter-laboratory variability in experimental protocols. An additional limitation is the lack of globally standardized quality-control and calibration procedures tailored to preclinical imaging systems. Emerging multimodal imaging platforms and high-fidelity disease models, such as genetically engineered rodents, large animals, and zebrafish, continue to enhance reproducibility, biological relevance, and translational potential. This review summarizes the development, capabilities, and limitations of preclinical PET and SPECT imaging, highlighting their expanding role in advancing molecular diagnostics, radiopharmaceutical development, and translational medicine in both preclinical studies and early-phase clinical research.

## 1. Introduction

Nuclear medicine is one of the youngest and most rapidly evolving fields of medicine. Its existence is owed to major discoveries in physics and chemistry at the turn of the 19th and 20th centuries. Research conducted by Marie and Pierre Curie, starting in 1896 on various uranium salt samples, confirmed the existence of elements exhibiting significantly higher radioactivity than uranium. In 1898, they published their findings announcing the discovery of polonium and radium, which laid the foundation for the development of nuclear medicine and the use of radioactivity in diagnostics and therapy [1,2,3]. Over the following decades, the physical principles of nuclear medicine became increasingly integrated with advances in molecular biology and genetics, leading to a more comprehensive understanding of biological processes at the molecular and systemic levels. In 2001 and 2002, groundbreaking studies were published describing the sequencing of human and mouse genomes and their comparison, contributing to a better understanding of the genetic basis of diseases [4,5]. This milestone in genomics enabled the development of personalized medicine, biomarkers, and preclinical biological studies aimed at improving clinical medicine.

The integration of molecular biology, genomics, and physical principles of nuclear medicine has also provided the conceptual and methodological basis for the development of small-animal imaging techniques. These approaches combine molecular-level investigation with radiochemical methods, making them indispensable tools for studying disease mechanisms, evaluating novel therapies, and bridging preclinical and clinical research, thus advancing translational medicine. Drivers for the advancement of small animal imaging techniques emerged, among others, from positions adopted by two organizations: the National Institute of Biomedical Imaging and Bioengineering (NIBIB), part of the National Institutes of Health (NIH), and the Association for Assessment and Accreditation of Laboratory Animal Care (AAALAC). Both institutions recognized that advances in imaging technology allow for less invasive research methods, which not only enhance the scientific value of experiments but also improve the welfare of laboratory animals, in line with ethical and regulatory requirements [6,7]. In preclinical research, mice and rats are the most commonly used animal species due to their small size, short life cycle, low maintenance cost, and reproductive efficiency [8,9]. The advantage of rodents as animal models also lies in their anatomical, physiological, and genetic similarity to humans, making them particularly valuable for translational studies [10].

Preclinical animal studies play a crucial role in the development of new therapies, although their translational value is sometimes limited. This issue can be addressed by improving model selection, optimizing study design, and enhancing methodological transparency including the use of randomization, blinding, and appropriate sample size calculation. Equally important is the implementation of minimal methodological standards and quality control (QC) procedures, which increase the reliability of obtained results and facilitate their translation into clinical practice [11,12]. An example of effective standardization that enhances reproducibility and robustness of findings is the study demonstrating that rapamycin extends lifespan in mice. This effect was independently replicated across three different research centers [13]. Another example of good translational practice is the use of biomarkers such as amyloid beta (Aβ) and phosphorylated tau protein in cerebrospinal fluid, which have been associated with Alzheimer’s disease progression in both clinical and preclinical settings [14].

Despite advances in preclinical in vivo imaging studies, post-mortem investigations remain essential for validating imaging biomarkers, elucidating disease mechanisms, and conducting detailed histological and molecular analyses that cannot be achieved otherwise. Although post-mortem analyses particularly in neuropathological research, continue to provide valuable insights into terminal disease alterations, their utility in evaluating therapeutic intervention efficacy is limited. Consequently, there is an increasing emphasis on functional readouts and in vivo monitorable biomarkers that more accurately reflect dynamic pathophysiological processes.

The most advanced imaging techniques, such as single-photon emission computed tomography (SPECT), positron emission tomography (PET), computed tomography (CT), and magnetic resonance imaging (MRI), have enabled non-invasive imaging of anatomical and functional structures in laboratory animals. Multimodal imaging systems (e.g., PET/CT, PET/MRI) are increasingly used, allowing for more precise and comprehensive analysis of biological processes. Particularly in the early 21st century, rapid development of hardware and advanced reconstruction and data analysis software in SPECT and PET contributed to significant progress in preclinical imaging of small animals, mainly rodents. Due to their small size, rodents require imaging systems with high spatial resolution, especially for studies involving complex structures such as the brain or heart. Accurate mapping of metabolic and receptor activity in such small organs demands high sensitivity and spatial resolution of detectors, which necessitates the use of specialized technological solutions such as monolithic detectors and advanced image reconstruction algorithms.

Currently, a wide range of animal models faithfully recapitulate human pathologies, including cardiovascular, autoimmune, oncological, and neurological diseases [15]. Only in 2018, approximately 12 million animals were used for research purposes within the European Union, highlighting the scale and importance of preclinical studies [16]. At the same time, initiatives aimed at reducing animal numbers through the development of alternative research methods are being pursued [17]. Increasingly, in silico approaches employing advanced computational models are also being implemented to predict the toxicity and biological properties of chemical compounds, thereby supporting efforts to minimize the use of animals in research [18].

The importance of preclinical studies continues to grow, supported by the rapid development of the pharmaceutical and biotechnology industries. Animal models enable the investigation of disease pathophysiology, drug development, and assessment of therapy efficacy and safety before clinical trials.

This review provides a comprehensive and up-to-date synthesis of the development and current applications of preclinical molecular imaging, with particular emphasis on PET and SPECT. It also discusses the role of animal models in molecular imaging, radiopharmaceutical development, and the evaluation of innovative therapeutic strategies. Its scientific value lies in integrating technological, methodological, and translational perspectives—from the historical foundations of imaging systems to detailed analyses of factors influencing data quality and modern multimodal solutions. The manuscript highlights how advancements in detectors, collimators, reconstruction algorithms, and quality-control procedures have markedly improved the sensitivity, spatial resolution, and quantitative reliability of small-animal imaging. It also underscores the critical impact of experimental conditions and protocol standardization on reproducibility and translational relevance. By combining discussions of animal models, hybrid imaging platforms, QC requirements, and the 3Rs framework, the article offers a coherent, practical, and clinically meaningful overview that supports the design, interpretation, and standardization of preclinical molecular imaging studies.

## 2. Fundamentals of Preclinical PET and SPECT

### 2.1. Principles of Nuclear Imaging

PET is based on imaging radionuclides, which emit positrons (β+) when they decay. The released positrons, interacting with nearby electrons, undergo annihilation. As a result of this process, they release energy in the form of two gamma-ray photons, which are emitted in opposite directions and at opposite angles. PET cameras are constructed with a ring of detectors for the detection of gamma radiation corresponding to an energy of 511 keV [19].

SPECT imaging uses radionuclides that emit gamma radiation photons during their radioactive decay. The emitted gamma rays must have specific energy levels, ranging from 100 keV to 200 keV [20]. SPECT is performed by rotating a gamma camera during the examination, which captures the gamma radiation emission in three dimensions.

The development of imaging equipment plays a key role in research using radiopharmaceuticals.

### 2.2. Radionuclides and Radiopharmaceuticals Used in PET and SPECT

Physical half-life is an important characteristic of radionuclides. In imaging studies, their values range from just a few hours to several days. Therefore, the half-life must be long enough to allow imaging in a timely manner after injection, but it should not be too long, as this unnecessarily increases the radiation dose during the examination [21].

PET employs short-lived positron-emitting isotopes, such as ^11^C (t_1/2_ = 20.4 min) and ^18^F (t_1/2_ = 109.7 min), to investigate biological processes in vivo, most commonly using the radiolabeled glucose analog ^18^F-fluorodeoxyglucose ([^18^F]FDG) [22,23].

SPECT imaging is unique in that it enables the relatively straightforward visualization of ligands, such as peptides or antibodies, using radionuclides including ^99m^Tc, ^111^In, or iodine isotopes (^123^I, ^125^I). Owing to their comparatively long physical half-lives (relative to most PET tracers), these radionuclides facilitate the assessment of slow kinetic processes and the simultaneous investigation of multiple molecular pathways [22].

Radionuclides form the physical foundation of PET and SPECT, while radiopharmaceuticals, created by coupling these radionuclides to biologically active molecules, enable targeted visualization and quantification of specific physiological and molecular processes in vivo. More detailed information on PET and SPECT radiopharmaceuticals, including their applications and key biological parameters, is summarized in Table 1.

### 2.3. Emerging Theranostic Radionuclide Platforms for Preclinical Imaging and Therapy

In addition to classical imaging isotopes, increasing interest is focused on element-matched theranostic radionuclide pairs. These pairs enable the use of chemically identical radiopharmaceuticals for both diagnosis and therapy. The concept of theranostics in nuclear medicine was first introduced in the treatment of thyroid cancer using radioiodine. This approach was described by Samuel Hertz and A. B. Roberts in 1946 and included both diagnostic and therapeutic applications with the same isotope [26]

Scandium represents one of the best characterized and most promising theranostic platforms. Due to the availability of chemically identical matched radionuclides, ^43^Sc and ^44^Sc are used for PET imaging, while ^47^Sc, a β emitter with concomitant γ emission, is suitable for radionuclide therapy and SPECT imaging. The longer half-lives of ^43^Sc/^44^Sc compared with ^68^Ga facilitate radiopharmaceutical transport and allow for extended imaging protocols. Stable chelation of scandium by DOTA (1,4,7,10-tetraazacyclododecane-1,4,7,10-tetraacetic acid) enables the use of the same ligands for both diagnostic and therapeutic applications [27]. Terbium is unique in nuclear medicine because it offers isotopes covering all four major imaging and therapeutic modalities. ^152^Tb is a positron emitter for PET, while ^155^Tb emits γ radiation suitable for SPECT. ^161^Tb combines low-energy β emission with γ radiation and Auger electrons, enhancing its therapeutic potential. ^149^Tb is an α emitter enabling targeted alpha therapy. Stable coordination of terbium with DOTA is consistent with theranostic principles. In a proof-of-concept study, four matched terbium isotopes were labeled with a folate derivative targeting folate receptors in mouse tumor models. The results demonstrated high and specific tumor uptake, effective PET and SPECT imaging, and significant tumor growth inhibition following α and β therapy. Despite these promising results, production and separation of terbium isotopes remain major challenges for clinical translation [28]. Another example is the theranostic pair ^203^Pb/^212^Pb, which combines SPECT imaging (^203^Pb) with α therapy (^212^Pb). This approach enables personalized, targeted anticancer treatment. Preclinical studies have shown that ^203^Pb accurately predicts radiopharmaceutical tumor distribution, while ^212^Pb therapy effectively inhibits tumor growth [29]. The isotope ^124^I is a long-lived positron emitter used in PET imaging for quantitative assessment of radiopharmaceutical distribution. ^211^At is a promising α emitter for targeted therapy, but its short half-life and logistical constraints limit direct PET imaging. Due to the chemical similarity between iodine and astatine, ^124^I can serve as a surrogate imaging isotope for ^211^At. This strategy enables optimization of therapy planning and dosimetry assessment in preclinical studies. Both isotopes can be incorporated into small organic molecules, facilitating the design of specific theranostic radiopharmaceuticals [30]. Finally, the therapeutic potential of the α emitter ^213^Bi has been evaluated in radioimmunoconjugates with bevacizumab. In parallel, the longer-lived bismuth isotopes ^205^Bi/^206^Bi were used to investigate pharmacokinetics and in vivo biodistribution. ^213^Bi-bevacizumab therapy significantly reduced tumor area compared with bevacizumab alone, confirming its therapeutic efficacy. The direct clinical application of the ^206^Bi/^213^Bi pair is not yet well established. However, preclinical studies indicate its potential for the development of safe and effective alpha-particle therapies [31].

Element-matched theranostic radionuclide pairs enable the use of chemically identical radiopharmaceuticals for diagnosis and therapy, with preclinical studies of scandium, terbium, lead, iodine/astatine, and bismuth isotopes demonstrating strong potential for personalized imaging-guided radionuclide treatment despite ongoing production and translation challenges.

### 2.4. Physical Determinants of Image Formation

To determine the origin of photons in SPECT, various collimators are used, which preferentially transmit parallel rays. Depending on their size, images with different spatial resolutions are obtained [32,33]. The design of collimators for clinical and preclinical applications has been extensively reviewed in the literature [34,35]. Collimators are typically made of high-density, high-atomic-number materials such as lead, tungsten, gold, or platinum. They feature a system of apertures that act as a filter, permitting only photons traveling along predetermined directions to reach the detector, where they are converted into an electronic signal and subsequently reconstructed into an image [36].

The sensitivity of a collimator can be increased by enlarging the aperture diameter; however, this typically results in reduced spatial resolution. To optimize the trade-off between sensitivity and spatial resolution, various design and technological strategies have been developed [36].

### 2.5. Biological Considerations in Tracer Use

Glucose metabolism is markedly increased in over 80% of tumor lesions, allowing PET to identify regions of elevated tracer uptake. A large meta-analysis comprising more than 18,000 patients demonstrated that [^18^F]FDG-PET achieves a sensitivity of 84% and a specificity of 88% (based on over 14,000 patient studies) for tumor detection [37]. In addition, [^18^F]FDG is extensively applied in functional brain imaging, where it serves as a proxy for neuronal glucose consumption, providing critical insights for the diagnosis of neurodegenerative disorders, epilepsy, and psychiatric conditions [38]. Different radiotracers rely on distinct mechanisms of uptake and clearance, which determine how they reflect underlying biological processes. [^18^F]FDG accumulates through metabolic trapping after phosphorylation, whereas receptor-targeted tracers such as [^68^Ga]Ga-DOTATATE or [^68^Ga]Ga-PSMA-11 undergo receptor binding followed by internalization. Perfusion tracers (e.g., [^99m^Tc]MIBI) depend on blood flow and mitochondrial membrane potential [22,23]. Understanding these mechanistic differences is essential for correct interpretation of PET and SPECT images. Beyond glucose metabolism, radiotracer uptake is strongly influenced by physiological and experimental factors including blood glucose levels, anesthesia type, body temperature, inflammatory status, and stress. Such variables can substantially alter biodistribution patterns; e.g., isoflurane decreases FDG uptake in brown adipose tissue and skeletal muscle, whereas ketamine/xylazine induces hyperglycemia, reducing tracer uptake in target tissues [39]. Rigorous control of such factors is therefore critical for ensuring data reproducibility and accurate biological interpretation [40].

Several biological phenomena may complicate data interpretation, including FDG uptake in inflammatory tissue, tumor heterogeneity leading to non-uniform tracer accumulation, and treatment-related metabolic changes such as steroid-induced hyperglycemia or immunotherapy-associated immune activation. These factors highlight the need to integrate PET/SPECT data with complementary biological or anatomical information [39]. FDG, despite its wide applicability, has important biological limitations. Tumors with inherently low glucose metabolism—such as certain prostate or neuroendocrine tumors—are poorly visualized with FDG, and the high physiological uptake of glucose in the brain complicates neuroimaging studies [37,38]. These limitations underscore the importance of alternative metabolic, receptor-based, or proliferation tracers tailored to specific disease contexts [22].

Additionally, tracer kinetics play a fundamental role in data interpretation. Time-activity curves, compartment-based models, and appropriate acquisition windows are necessary to quantify biological processes accurately. Identical standardized uptake values (SUV) may represent different biological states depending on the tracer’s kinetic profile, organ system, or physiological conditions [21].

Collectively, these biological considerations emphasize that accurate PET and SPECT imaging requires not only high-performance instrumentation but also careful control of experimental variables, a thorough understanding of tracer mechanisms, and appropriate kinetic modeling. These factors are essential for achieving reliable, reproducible, and biologically meaningful molecular imaging results [39,41].

## 3. Preclinical PET and SPECT Instrumentation and Technological Development

With the growing demand for in vivo monitoring of biological processes in animal models, molecular imaging techniques have undergone significant advancements, including PET and SPECT. While classical approaches, such as autoradiography or quantitative measurement of radioisotopes in post-mortem tissue, have continued to provide valuable insights, these methods are inherently limited in their ability to capture dynamic, whole-body processes [42,43].

### 3.1. Evolution of Preclinical PET Technology

PET imaging would not have been possible without key discoveries in chemistry, physics, and physiology/medicine. The first of these was the discovery of the positron by C. D. Anderson in 1932. This finding linked Dirac’s theory of antimatter with the basic principles of β^+^ radioactive decay. Positrons annihilate with electrons, producing photons with an energy of 511 keV. These photons are detected and used in PET imaging [44].

Another major milestone was the invention of the cyclotron by Ernest Orlando Lawrence in 1929. The cyclotron enabled the production of medically relevant radioisotopes, including ^11^C, ^13^N, ^15^O, and ^18^F, which emit β^+^ particles [44]. Initially, not all isotopes were readily available. Their production expanded over subsequent decades with improvements in cyclotron technology. In 1934, György Hevesy first used the artificial radioisotope ^32^P to study bone growth in rats [45]. For his pioneering work on isotopes as tracers in biological research, Hevesy received the Nobel Prize in Chemistry in 1943 [46]. His method became the foundation for modern radiotracers used in PET.

Advances in radiation detectors and image reconstruction were also critical. In 1948, Robert Hofstadter built a scintillation counter for ionizing radiation [47]. The development of practical PET detectors is primarily credited to Michel Ter-Pogossian [48]. The mathematical basis for PET image reconstruction is the Radon transform, formulated in 1917. It allows the spatial distribution of radioisotope activity in the body to be reconstructed [49].

Ter-Pogossian received numerous awards for his pioneering work. He contributed to the medical application of the cyclotron, studies with ^15^O tracers, kinetic analysis, and the development of PET systems. In 1967, he published a textbook on diagnostic radiology [48].

The first dedicated preclinical PET scanners emerged in the 1990s, built on solutions originally developed for clinical imaging [50,51,52]. For example, the SHR-2000 system developed by Hamamatsu employed bismuth germanate detectors arranged in four rings of 384 mm diameter, enabling rat brain imaging with spatial resolution in two directions [53]. Subsequently, other systems were developed, including RATPET (based on 16 BGO detectors and photomultipliers) offering ~2.2 mm resolution and the Sherbrooke APD PET (using avalanche photodiodes and a wobbling technique), which improved resolution to 2.1 mm [54]. A further milestone was the market introduction by Concorde Microsystems Inc. of the microPET series dedicated to imaging primates and rodents. The first-generation systems (P4 and R4) used lutetium oxyorthosilicate (LSO) scintillators in four rings; later models (Focus-120, Focus-220 and Inveon-DPET) offered improved detector geometry, enhanced electronics and greater sensitivity and field-of-view (FOV). In particular, the Inveon model stood out for its largest axial extent and roughly three-fold higher sensitivity compared to predecessors, as well as its tri-modal capability [55,56]. Universities in Ferrara and Pisa developed a unique scanner YAP-(S)PET, combining PET and SPECT techniques in a single gantry. The system featured four rotating detector heads with YAP crystals set opposite one another at a 150 mm distance. Its unique design allowed rapid switching between PET and SPECT modes via collimator exchange. However, due to the large solid angle of the detectors, the system suffered from significant signal pile-up, necessitating modernization of the electronics, which significantly improved acquisition parameters [57,58]. Other preclinical systems include the Mosaic-HP by Philips (using LYSO crystals for whole-body rodent imaging in a single position) and ClearPET with adjustable rotating detector heads of LYSO/LuYAP [59,60]. The FLEX Triumph, as the first commercial tri-modality PET/SPECT/CT system, combined CZT detectors with PET subsystems based on BGO/PMT, offering a large FOV and high sensitivity [61]. The LabPET series from Gamma Medica/GE Healthcare features phoswich detectors (LYSO/LGSO) and avalanche photodiodes (APD); their design minimizes parallax error [62,63,64]. SEDECAL introduced systems such as Argus and VrPET/CT with very fine crystal elements, resulting in high spatial resolution albeit sometimes at the expense of sensitivity.

The next generation of preclinical PET scanners from companies such as Mediso, Bruker Oncovision, and Inviscan represents a new level of technological advancement. These systems offer improved spatial resolution, sensitivity, acquisition time, and multimodality integration. In the design of current systems we increasingly see finer detector elements (e.g., monolithic LYSO crystals or multilayer depth-of-interaction (DOI) detectors), which translate into better anatomical detail and more uniform resolution across the field of view. Moreover, a broad range of multimodal solutions such as PET/CT, PET/MR, PET/SPECT/CT or PET/MR/SPECT enable comprehensive functional-structural studies within a single system, eliminating the need to transfer the animal between devices. The FOV allows imaging of the whole body of mice and rats. Rotating gantries, modern photomultiplier fabrication and the ability to perform both dynamic and static imaging modes make these systems extremely versatile. Concurrently, improved system performance in terms of sensitivity and contrast at lower radiation dose enhances experimental safety and repeatability [54].

### 3.2. Physical Determinants of PET Resolution and Accuracy in Multimodal Imaging

Although spatial resolution in PET is to a large extent determined by scanner construction, physical processes occurring in the subject itself also have a significant influence. One of the key limitations is the distance traveled by the positron before annihilation with an electron. This range depends on the type of radionuclide (e.g., 0.5 mm for ^18^F, and up to 4 mm for ^82^Rb) and the medium through which the positron travels (for example, air or tissue). This effect may substantially degrade the effective spatial resolution, especially in preclinical studies where structures are small. In high-resolution small scanners, even a small positron range may reduce image quality by tens of percent. For high-energy positron-emitting isotopes (e.g., ^82^Rb), their use in imaging small structures such as a rodent heart may be impractical. Therefore, appropriate corrections or the selection of short-range isotopes is necessary, particularly in studies requiring high spatial precision [65,66]. Other physical factors that affect spatial resolution in PET include the non-collinearity of annihilation photons, as well as photon scatter and attenuation within tissue. Non-collinearity leads to deviations of annihilation photons from the ideal 180° trajectory, resulting in inaccurate line-of-response (LOR) determination and degradation of spatial resolution, particularly in scanners with a large detector ring diameter [67,68]. Scatter and attenuation reduce image contrast and may introduce errors in reconstruction if not properly corrected [69,70]. Although attenuation is less significant in small animals (such as a single mouse), its influence becomes noticeable when imaging multiple animals or larger objects [71].

Accurate spatial alignment between functional and anatomical images is critical for reliable preclinical imaging. Chow et al. [72] developed a hardware-software approach for multimodal image registration in small animals, combining a dedicated imaging chamber, a 3D mesh phantom, and a 15-parameter transformation matrix. Compared with conventional 6-parameter rigid-body models, this framework substantially improved registration accuracy, achieving an average spatial precision of 0.335 mm in both phantom and in vivo studies. Complementing this, Vandervoort et al. [73] applied Monte Carlo modeling to single-mode transmission data in small-animal PET scanners. The approach enabled detailed simulation of photon attenuation and scatter within biological tissues, improving quantitative accuracy. By accounting for these physical effects, the model enhances both the reliability of reconstructed images and the accuracy of activity measurements. Together, these studies highlight that integrating robust registration techniques with physics-based corrections, supported by advanced software and dedicated hardware, can overcome spatial and quantitative limitations in preclinical imaging. Ultimately, these innovations provide a framework for more reliable and reproducible assessment of molecular targets in small-animal studies.

### 3.3. Evolution of Preclinical SPECT Technology

SPECT remains a widely utilized imaging modality in both clinical diagnostics and preclinical research. Compared to PET, SPECT offers certain technological advantages due to its direct detection of gamma photons emitted by radionuclides, eliminating the intermediate steps of positron emission and annihilation inherent in PET. These processes can reduce spatial resolution in PET, thus favoring SPECT systems in specific applications. Moreover, SPECT allows for simultaneous imaging with multiple radiotracers labeled with isotopes emitting gamma photons of distinct energies, greatly enhancing the modality’s flexibility in experimental design and tracer selection [74,75,76].

Early preclinical SPECT systems typically employed single-pinhole collimators and a single gamma camera. The performance of these collimators defined by parameters such as pinhole diameter, collimation angle, and the attenuation coefficient of the collimator material directly influenced resolution and sensitivity [36]. Large-aperture collimators (e.g., 3 mm) often yielded sensitivities below 1%, necessitating prolonged acquisition times (up to an hour) and higher radiation doses to animals [77]. Modern systems have addressed these issues with multipinhole collimators, which significantly enhance the resolution-to-sensitivity ratio [78]. By simultaneously utilizing more of the detector area, these systems enable shorter scan durations, reduced dose burdens and improved visualization of small pathological structures, such as microlesions or inflammation [79]. Moreover, object magnification based on the pinhole–object–detector geometry can be leveraged to enhance effective spatial resolution [36].

### 3.4. Physical and Methodological Factors Affecting Quantification in Preclinical SPECT

Despite its advantages, one of the main challenges in preclinical SPECT, particularly in small animal studies is its limited quantitative accuracy [80]. While this modality enables dynamic assessment of metabolic changes in specific organs and monitoring of therapeutic responses, such as tumor regression, quantitative reliability can be affected by several physical and technological factors. These include photon attenuation and scatter within biological tissues, which degrade the image, as well as the partial volume effect (PVE), which becomes particularly relevant when the size of the target structure approaches the system’s intrinsic spatial resolution, leading to activity underestimation and image blurring [78,80]. Additionally, the limited number of detected counts affects image quality and temporal resolution, both critical parameters in dynamic imaging studies [81].

In longitudinal studies, especially in oncology models, PVE can substantially compromise quantitative analysis due to dynamic changes in tumor size. Various correction strategies can be applied in practice to mitigate this effect and improve quantitative accuracy [82,83].

Accurate quantitative representation of in vivo radiotracer distributions in preclinical SPECT remains a central technical objective, requiring sophisticated correction and reconstruction methodologies. Modern approaches focus on mitigating the physical and statistical limitations that degrade image quality, including photon attenuation, scatter, and the PVE.

Attenuation and scatter correction are now routinely applied in state-of-the-art preclinical SPECT systems. These methods compensate for photon loss and energy degradation occurring within biological tissues. Empirical approaches, such as Chang’s method, originally developed for clinical SPECT, have been successfully adapted for small-animal imaging [84]. They have been shown to significantly improve quantitative accuracy, particularly in regions with high density differences, such as the thoracic or abdominal cavities [85,86].

Advanced scatter compensation techniques, including dual-energy window and Monte Carlo–based methods, further enhance image fidelity. They account for energy spectrum distortions and spatially varying scatter components. Phantom and animal studies have demonstrated that these approaches improve both visual contrast and quantitative recovery [86].

Resolution recovery and partial volume correction represent another essential component of quantitative improvement. Given that the effective spatial resolution in small-animal SPECT (≈1 mm or less) often approaches the size of the structures under study, PVE becomes a dominant source of error, leading to activity underestimation in small lesions or organ subregions. Recent studies have reviewed and benchmarked multiple PVE correction strategies ranging from geometric transfer matrix (GTM) modeling to deconvolution-based restoration highlighting their role in restoring true tracer concentration without amplifying noise [87]. These methods, when integrated into iterative reconstruction pipelines, substantially enhance spatial accuracy and the detectability of fine structures.

Iterative image reconstruction algorithms particularly ordered subsets expectation maximization and its extensions, remain the cornerstone of modern quantitative SPECT. Unlike analytical techniques (e.g., filtered back-projection), iterative models can incorporate accurate system response functions, collimator–detector geometry and physical phenomena such as parallax and pinhole penetration. Recent implementations in multipinhole SPECT systems have demonstrated the capacity to achieve absolute quantification, with recovery coefficients approaching unity in well-calibrated systems [80,88]. Hybrid reconstruction schemes that couple resolution recovery with statistical noise modeling further contribute to maintaining quantitative integrity, even under low-count conditions typical of longitudinal or dynamic studies [88].

A major breakthrough in SPECT technology was the development of integrated SPECT/CT systems, where in both imaging components are mounted on a single gantry, enabling co-registered, automated data acquisition. This configuration facilitates accurate spatial registration between functional SPECT data and anatomical CT images, eliminating misalignment artifacts due to repositioning. CT images serve not only for anatomical localization but also for attenuation correction and collimator-response modeling based on depth-dependent sensitivity. Seo et al. demonstrated that these corrections significantly improved both qualitative and quantitative image accuracy, particularly for high-energy photon emitters and small target structures [89,90]. In preclinical studies involving small and dynamic regions of interest, integrated SPECT/CT systems improve the reliability, reproducibility, and accuracy of the resulting data. Today, these hybrid scanners are essential tools in molecular imaging and precision translational research.

### 3.5. Advances in Multimodal, Hybrid, and Quad-Modality Imaging Platforms in Preclinical PET and SPECT

Over the past decades, preclinical PET and SPECT systems have undergone substantial technological advancements aimed at improving spatial resolution, sensitivity, and quantitative accuracy parameters critical for high-quality molecular imaging. Early systems were limited in their ability to resolve small anatomical structures or detect low-abundance molecular targets. Advances in detector materials, collimator designs (e.g., multipinhole collimators in SPECT), and radiotracer chemistry have gradually expanded the capabilities of these modalities, enabling more precise and reliable imaging. Hybrid imaging systems integrate functional and anatomical information, combining the high sensitivity of nuclear imaging with the high spatial resolution of CT or the superior soft-tissue contrast of MRI.

Furthermore, the emergence of modern multimodal platforms continues to push the boundaries of preclinical imaging. In 2024, Wang et al. [91] described an innovative quad-modality system integrating PET, SPECT, spectral CT, and cone-beam CT, whose performance was evaluated using Monte Carlo simulations. This platform allows highly precise imaging by combining functional and anatomical information from multiple modalities, which is particularly valuable for planning and monitoring complex therapies such as radiotherapy [92]. Collectively, these technological innovations from advanced detectors and collimators to hybrid and multimodal systems have substantially enhanced the translational potential of preclinical nuclear imaging. They provide researchers with more comprehensive tools to study molecular processes, optimize therapeutic protocols, and facilitate the translation of findings from preclinical models to clinical applications.

Preclinical PET and SPECT technologies have evolved dramatically, driven by advances in detector design, radiotracer production, image reconstruction, and multimodal integration. These innovations have greatly improved spatial resolution, sensitivity, and quantitative accuracy, enabling more precise and comprehensive imaging of biological processes in small animals. As a result, modern hybrid and multimodal platforms now offer powerful tools that enhance experimental reliability and accelerate translation from preclinical research to clinical applications. The timeline and key milestones in the development of imaging techniques, from the earliest discoveries to the present day, are presented in Figure 1.

## 4. Applications of Noninvasive Imaging in Small-Animal Models

### 4.1. Role of Small-Animal Models in Biomedical Research

The August Krogh Principle, formulated as “for many problems, there is an animal on which it can be most conveniently studied”, was elaborated in detail by Hans Adolf Krebs in his article published in the Journal of Experimental Zoology in 1975 [93]. This principle is particularly relevant to the application of noninvasive imaging techniques in small-animal research. Krebs emphasized that the selection of an appropriate animal model is crucial for effectively addressing specific biomedical questions. In the context of imaging preclinical research, such as PET and SPECT, the use of animal models represents a fundamental tool for investigating the pathophysiology of various diseases and assessing therapeutic efficacy, representing an essential step toward translating findings into the clinical setting in humans.

The application of noninvasive small-animal imaging has significantly advanced preclinical research. Ethical limitations on human experimentation, together with the inability to fully reproduce complex disease mechanisms under in vitro conditions, support the use of appropriate in vivo animal models when designing experimental strategies [76].

### 4.2. Longitudinal Imaging and Ethical Advantages

In vivo imaging techniques play a key role in increasing the efficiency of preclinical research, as they enable repeated, noninvasive measurements in the same subject without the need for euthanasia [94]. This approach allows not only longitudinal monitoring of biological processes but also a substantial reduction in the number of animals required for experiments, since each animal can serve as its own control [51]. These practices fully comply with the ethical 3R principles (Reduction, Refinement, Replacement) established by Russell and Burch, which form the foundation of responsible research involving laboratory animals [95,96].

### 4.3. Translational Relevance and Model Selection

All these considerations are consistent with the definition of small-animal imaging proposed by Zanzonico, who described it as “a method for the biological assessment of structure and function in vivo using noninvasive means, enabling the collection of quantitative data in both health and disease” [97].

Animal studies provide the most comprehensive insight into biological processes [98]. Complex physiological and pathological mechanisms cannot be fully characterized using cell cultures or isolated tissue samples alone. Only within a living organism is it possible to observe and analyze the dynamic and integrated interactions among the immune, endocrine, and nervous systems [99]. These systems function in constant communication and interdependence, which is essential for understanding complex biological networks and for predicting disease onset, progression, and treatment responses under specific conditions [100]. Therefore, in vivo models remain indispensable in biomedical research, particularly in translational studies aiming to bridge basic science with clinical applications [101].

Small animals, mainly rodents such as rats and mice, are the most commonly used species in animal models. Mice are particularly suitable for cancer studies and for us in commercially available a transgenic models [102], whereas rats are preferred for studies involving long-term anesthesia due to their relatively high tolerance [103]. The use of inbred strains may further contribute to reducing the total number of animals required in an experiment [104]. Despite their advantages, rodent models have certain limitations, including small organ size and limited blood volume, which may hinder precise imaging with noninvasive methods [11]. However, over the past two decades, there has been rapid technological progress in imaging technologies systems dedicated to small animals, such as PET, SPECT, CT or MRI enabling the overcoming of these limitations. Modern scanners now provide high spatial resolution in both axial and transverse planes, allowing detailed imaging of small structures within the rodent brain or other organs. Moreover, the high sensitivity at the center of the field of view enables the detection of subtle metabolic and functional changes. Owing to these technological advancements, microscanners have become increasingly essential tools for precise preclinical research using rodent models [105]

Similarly, PET or SPECT imaging of the mouse heart, aimed at achieving organ-level spatial resolution with temporal resolution comparable to human clinical studies, requires submillimeter spatial resolution and frame rates approximately ten times higher than those used in standard clinical protocols. Notably, scanners capable of meeting these demanding parameters are now available [106]. The fundamental principles of PET and SPETC imaging remain consistent across clinical and preclinical applications. In both cases, the primary objective is to obtain the highest possible image quality, which directly depends on maximizing the signal-to-noise ratio (SNR). Achieving this goal requires advanced detector technologies, precise data acquisition systems, and optimized reconstruction software to reduce artifacts and improve resolution [41].

### 4.4. Physiological Factors Influencing Imaging Outcomes

While high spatial and temporal resolution are key determinants of image quality, an equally important factor lies in the biological specificity of the imaging process. In anatomical and functional imaging, the use of radiopharmaceuticals or contrast agents is essential for visualizing and quantifying specific biological processes in vivo. These agents enable the detection of physiological and molecular changes by selectively binding to or accumulating in target tissues, thereby reflecting underlying biochemical activity. An equally in imaging research important aspect is the accurate localization and quantification of the administered imaging tracers within the target tissue or organ. These compounds should be used in the lowest possible dose that still provides sufficient contrast and detection sensitivity, minimizing the biological burden on the animal (or patient) and reducing the impact of radiation dose on imaging results [21].

Another critical challenge in preclinical imaging is ensuring the reliability and reproducibility of results. The implementation of standardized experimental protocols helps reduce variability and improves comparability across research centers. However, several factors can influence data variability, including the type and depth of anesthesia, physiological parameters of the animals, handling procedures and methods of image data analysis.

### 4.5. Representative Methodological Examples: FDG Biodistribution, Anesthetic Effects, and Dynamic Acquisition Strategies

As demonstrated by the following examples, the development of detailed experimental protocols and appropriate personnel training not only increases the accessibility of advanced imaging techniques but also promotes standardization, which is essential for ensuring result comparability among laboratories.

Fueger et al. [39] conducted a detailed analysis of how different experimental conditions specifically the type of anesthesia, body temperature and fasting duration prior to imaging affect the biodistribution of ^18^F-FDG in mice. They demonstrated that these factors can significantly alter the interpretation of quantitative data, particularly in studies of glucose metabolism. Isoflurane anesthesia was associated with reduced FDG uptake in brown adipose tissue (BAT) and skeletal muscles, while increasing uptake in the liver, kidneys, and myocardium. In contrast, ketamine/xylazine anesthesia induced hyperglycemia, leading to elevated blood glucose levels and decreased FDG uptake in target tissues. These alterations disrupt normal physiological organ-to-organ relationships and can substantially affect the accuracy of quantitative PET analysis [39].

Other studies [44] have confirmed that small-animal PET imaging can be performed repeatedly in longitudinal studies without compromising the physiological condition of the animals, while maintaining high image quality and reproducibility of quantitative data. These experiments utilized microPET-R4 and microPET-FOCUS F-220 systems, and the developed protocols enabled effective monitoring of biological changes over time using a wide range of radiotracers labeled with ^11^C, ^15^O, ^18^F, ^64^Cu, and ^86^Y [40].

Finally, Cunha et al. [107] suggested that issues related to data reliability and result variability can be mitigated through the use of phantom studies, which enable the correlation of imaging parameters, QC of the system, and improvement of spatial fidelity. Such procedures contribute to obtaining reliable and reproducible imaging data essential for robust quantitative analysis [107].

Noninvasive imaging in small-animal models enables longitudinal, ethically responsible studies that capture complex biological processes in vivo. Advances in PET, SPECT, CT, and MRI allow high-resolution, quantitative assessment of disease mechanisms and treatment responses in rodents. Standardized protocols, controlled physiological conditions, and quality-control tools ensure reliability and translational relevance of imaging results. A point-to-point advisory plan for using noninvasive imaging in small-animal research is presented in Figure 2.

## 5. Challenges, Limitations and Benefits in Preclinical Imaging Studies

### 5.1. Limitations of Preclinical Imaging Studies

Preclinical studies form the cornerstone of the development of novel therapies and diagnostic methods; however, their nature and limitations differ substantially from those of clinical research. This section presents the key challenges associated with conducting preclinical studies, with particular emphasis on molecular imaging.

One of the main challenges in preclinical research is the absence of a global regulatory body enforcing mandatory QC standards and standardized protocols for imaging equipment used in these studies. Consequently, uniform QC standards often do not exist even within individual institutes, leading to substantial difficulties in the reproducibility and comparability of results. Currently, only expert guidelines are available, such as the Preclinical SPECT and PET: Joint EANM and ESMI Procedure Guideline for Implementing an Efficient Quality Control Programme and the European Association of Nuclear Medicine (EANM) guidelines on good radiopharmaceutical practice (cGRPP) [108]. Moreover, manufacturers of preclinical imaging systems (e.g., Mediso, Siemens, GE, Bruker) provide proprietary QC protocols, which may or may not align with these recommendations.

In practice, the lack of unified and rigorous QC standards may result in serious consequences, as evidenced by studies demonstrating the impact of faulty or damaged detector modules in PET systems. Using a homogeneous 68Ge phantom as an experimental model, McDougald and Mannheim showed that even a single defect within a detector block can cause significant image distortions and deviations in SUVs of up to 15%. Moreover, the number of defective modules correlates directly with the severity and extent of the resulting artifacts. Different image reconstruction algorithms exhibit varying sensitivities to such defects, with 2D algorithms being more susceptible to errors than 3D approaches. Importantly, these malfunctions often remain undetected, increasing the risk of misinterpretation and compromising the reliability of quantitative PET analysis. In addition to the phantom experiments, the authors also used PET/CT images of a rat following [^18^F]FDG administration to illustrate the potential impact of such artifacts in the context of preclinical animal imaging [109].

A practical example of the consequences of inadequate QC is imaging of rats after myocardial infarction induction, where each procedure imposes physiological stress on the animal, including anesthesia, radiation exposure, and administration of radiotracers and contrast agents. In longitudinal studies, these procedures are repeated multiple times, making proper equipment function and high data acquisition sensitivity critical for obtaining reliable and reproducible results. Malfunctioning equipment leads not only to a loss of scientific value but also to unnecessary animal stress and mortality, which contradict ethical and scientific principles [109].

Preclinical studies are also characterized by irregularity due to intensive scanning sessions involving large numbers of animals, often sequentially or simultaneously, which may affect data consistency. The complexity of imaging protocols, necessitated by variables such as the specific physiology of animal models (e.g., differences in murine cardiac function) and the large volume of generated images, adds further challenges.

Furthermore, the high costs of purchasing and maintaining imaging equipment, infrastructure requirements, and limited access to advanced systems represent significant barriers to the advancement of preclinical imaging research. Equally important is the shortage of adequately trained personnel, including medical physicists responsible for QC supervision during experiments. Additionally, the need for numerous approvals and permits related to atomic law and animal protection regulations complicates and prolongs the research process.

### 5.2. Advantages, Model Diversity and Technological Advances in Preclinical Imaging

The lack of stringent regulatory constraints, combined with the wide diversity of animal models and tracers, affords users of preclinical PET and SPECT systems a markedly greater degree of flexibility in the design and modification of experimental protocols. This flexibility considerably exceeds that available in clinical research settings.

Importantly, preclinical studies in large animals, such as pigs [110] or non-human primates [111], provide a critical translational bridge between rodent models and clinical research. Large organs allow repeated physiological measurements and serial blood sampling, even during recovery periods. This feature enables a more accurate representation of human biological processes. Anatomical and physiological similarities to humans also permit the use of clinically relevant imaging techniques and therapeutic protocols. Maintaining large animals requires specialized infrastructure and entails higher costs. Nevertheless, their use substantially enhances the translational value of preclinical studies, bringing experimental findings closer to clinical conditions.

In addition to large-animal models, alternative vertebrate species are gaining attention. These organisms have evolved to thrive in specific environmental niches and offer novel insights into fundamental biological mechanisms. Studies involving amphibians, reptiles, and fish provide complementary perspectives on physiology, genetics, and disease progression. A particularly promising example is the development of an imaging platform for adult zebrafish (*Danio rerio*) maintained in a stable physiological state. This platform enables high-resolution molecular imaging using preclinical PET/CT systems. Notably, *Danio rerio* shares at least one orthologue with 82% of the 3176 human genes associated with disease. This further supports its translational relevance as a model organism in biomedical research [112,113]. Two independent studies show how technical limitations in small zebrafish models can be overcome. In one study, a high-sensitivity PET/CT system and a custom platform allowed stable immobilization of the fish in water. The zebrafish was placed at the center of the scanner’s field of view. This minimized signal degradation and improved detection. Very low activities of the [^18^F]FDG radiotracer were detectable. Hybrid PET/CT imaging enabled precise correlation of functional signals with anatomy. A 3D-printed zebrafish model was used to assess detection limits and validate imaging protocols. These results demonstrate that meaningful quantitative data can be obtained despite limited PET resolution [112]. In the second study, a digital zebrafish phantom proved essential. It enabled PET simulations and evaluation of aquatic effects on image quality. The phantom allowed testing tracer detectability in small structures, such as the brain and heart. Imaging protocols could be optimized without frequent in vivo experiments. Analyses showed that simplified water attenuation maps slightly degraded image quality [114].

Moreover, the development of CRISPR-Cas9 technology, which enables precise gene editing, has revolutionized the modeling of cancer in animal systems. This approach allows for the rapid generation of accurate tumor models, which can subsequently be monitored using molecular imaging techniques such as PET or SPECT. The combination of precise gene editing with advanced imaging modalities facilitates a deeper understanding of tumor biology and enables the evaluation of novel therapeutic strategies in vivo [115].

### 5.3. Technological Progress and Ethical Benefits in Preclinical Imaging

In conclusion, despite the aforementioned limitations, current preclinical imaging systems offer a range of significant advantages. The high spatial resolution of scanners allows for the integration of anatomical and functional imaging (e.g., PET/CT). Importantly, molecular imaging enables visualization without damaging biological tissues. Methodological advances also allow the use of a wide array of contrast agents, isotopes, and radiotracers, thereby enhancing both the scope and accuracy of preclinical studies [24,108].

It is noteworthy that the development of hybrid SPECT and PET systems coupled with CT or MRI provides flexible imaging options, as these modalities can be used either in combination or independently, depending on the study requirements. In the context of repeated imaging, it should be emphasized that CT involves a low radiation dose, whereas MRI is entirely non-ionizing. Moreover, many preclinical imaging techniques have clinical counterparts, facilitating the translation of findings from animal models to human applications [91,108].

Another important advantage is the possibility of performing multiple studies on a single animal in accordance with the 3Rs principle (Replacement, Reduction, Refinement), which reduces animal usage and improves the ethical standards of research. By demonstrating that biological processes could be monitored through radioactive markers rather than repeated tissue dissections, Hevesy introduced a methodological shift that anticipated what would later become the Reduction principle of the 3Rs [116] subsequently formalized by Russell and Burch in 1959 [96].

Preclinical imaging offers powerful tools for studying disease mechanisms, but its reliability is limited by the lack of unified quality-control standards, equipment variability, high operational costs, and the need for specialized personnel. At the same time, diverse animal models—from rodents to large animals and zebrafish—together with advances in gene editing and hybrid PET/SPECT/CT/MRI systems, greatly enhance the translational potential of preclinical research. Modern imaging technologies enable high-resolution, ethically aligned, and clinically relevant studies that support the development of new diagnostics and therapies. A conceptual framework summarizing these challenges and advantages is presented as a graphical overview in Figure 3.

### 5.4. Role of Artificial Intelligence in Preclinical PET and SPECT Imaging

Artificial intelligence (AI) is increasingly being applied in preclinical PET and SPECT imaging of small animals, offering significant improvements in image reconstruction, noise reduction, and spatial resolution. In addition, AI-based methods enable automated segmentation and advanced quantitative analysis of functional imaging data. Beyond technical enhancements, AI holds considerable translational potential in preclinical molecular imaging. By improving quantitative accuracy, facilitating protocol standardization, and enhancing reproducibility across laboratories, AI-driven approaches may substantially strengthen the reliability and comparability of preclinical PET and SPECT studies. Despite these promising developments, no EANM/ESMI guidelines currently exist regarding the implementation of AI in preclinical PET/SPECT imaging. Nevertheless, AI-based methodologies are expected to play an increasingly important role in the future, particularly in automated image analysis, multimodal data integration, and the optimization of imaging workflows.

## 6. Conclusions

Noninvasive molecular imaging of small animals plays an important role in modern biomedical research. By enabling longitudinal and quantitative assessment of biological processes in vivo, it provides functional and molecular insights that cannot be achieved with traditional histological or ex vivo methods. Importantly, preclinical imaging employs the same scanner modalities and imaging principles as those used in human studies, allowing experimental disease models to closely mimic clinical scenarios. This technological and methodological continuity enables the induction and monitoring of pathological processes in animals using the same PET, SPECT, CT, or MRI approaches applied in patients, thereby facilitating direct translation of experimental findings into clinical practice. These techniques enhance the predictive and translational value of preclinical models, supporting more accurate extrapolation of experimental results to human disease. In addition, they contribute to the ethical refinement of animal studies by reducing the number of animals required and minimizing biological variability. Although challenges related to instrumentation, protocol standardization and reproducibility across laboratories remain, ongoing advances in imaging technology continue to expand the predictive power and clinical relevance of preclinical molecular imaging.

## Figures and Tables

**Figure 1 cells-15-00073-f001:**
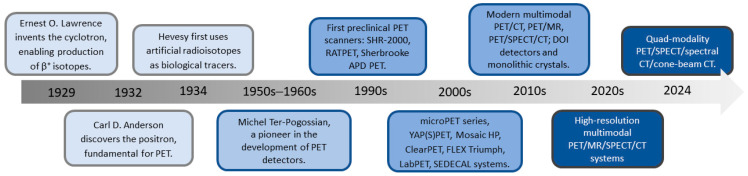
The timeline of imaging technique development, from first discoveries to current time. CT, computed tomography; MR, magnetic resonance imaging; PET, positron emission tomography; SPECT, single-photon emission computed tomography.

**Figure 2 cells-15-00073-f002:**
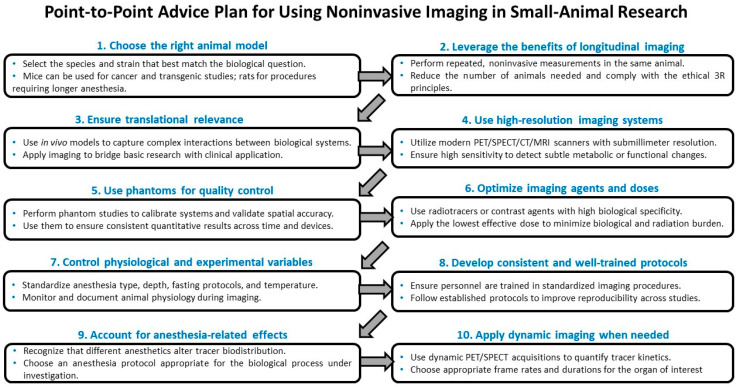
Summary of key recommendations for conducting noninvasive imaging studies, with practical guidance framework for optimizing PET/SPECT/CT/MRI imaging in small-animal research.

**Figure 3 cells-15-00073-f003:**
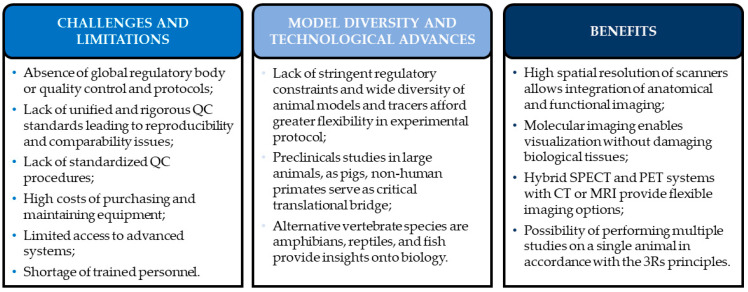
Infographic summarizing the challenges and limitations, model diversity, and technological advances and benefits of using preclinical small-animal imaging tools.

**Table 1 cells-15-00073-t001:** Overview of clinically used PET and SPECT radiopharmaceuticals: molecular targets, biological parameters and representative applications [24,25].

Radiopharmaceutical	Application	Parameter	PET	SPECT
[^18^F]FDG	Oncology, inflammation, neurology	Glucose uptake; GLUT; glycolysis	✓	
[^18^F]FCH/[^11^C]Choline	Prostate cancer	Membrane turnover	✓	
[^18^F]FLT	Proliferation	DNA synthesis	✓	
[^68^Ga]Ga-DOTATATE	Neuroendocrine tumors	SSTR2 density	✓	
[^68^Ga]Ga-PSMA-11	Prostate cancer	PSMA expression	✓	
[^18^F]NaF	Bone metastases	Osteoblastic activity	✓	
[^11^C]PIB	Alzheimer’s disease	Aβ	✓	
[^99m^Tc]MDP	Bone imaging	Osteoblastic turnover		✓
[^99m^Tc]MIBI	Cardiac perfusion	Mitochondrial potential		✓
[^201^Tl]TlCl	Myocardial perfusion	Na^+^/K^+^-ATPase		✓
[^123^I]MIBG	Neuroblastoma	NET uptake		✓
[^111^In]Octreotide	Neuroendocrine tumors	SSTR		✓
[^99m^Tc]HMPAO	Brain perfusion	CBF		✓
[^99m^Tc]MAG3	Renal imaging	Tubular secretion		✓
[^99m^Tc]DTPA	Renal filtration	GFR		✓

Aβ, amyloid beta protein; CBF, cerebral blood flow; DOTATATE, DOTA-Tyr^3^-octreotate; DNA, deoxyribonucleic acid; DTPA, diethylenetriamine pentaacetic acid; FDG, fluorodeoxyglucose; FCH, fluorocholine; FLT, fluorothymidine; GFR, glomerular filtration rate; GLUT, glucose transporter; HMPAO, hexamethylpropyleneamine oxime; MAG3, mercaptoacetyltriglycine; MDP, methylene diphosphonate; MIBG, metaiodobenzylguanidine; MIBI, methoxyisobutylisonitrile; Na^+^/K^+^-ATPase, sodium–potassium adenosine triphosphatase; NET, norepinephrine transporter; PET, positron emission tomography; PIB, Pittsburgh compound B; PSMA, prostate-specific membrane antigen; SPECT, single-photon emission computed tomography; SSTR, somatostatin receptor/somatostatin receptor; TlCl, thallium chloride.

## Data Availability

No new data were created or analyzed in this study. Data sharing is not applicable to this article.

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
