# Peer review of "Preclinical PET and SPECT Imaging in Small Animals: Technologies, Challenges and Translational Impact"

_cells, 2025, doi:10.3390/cells15010073_

Round 1
Reviewer 1 Report
Comments and Suggestions for Authors
The review summarized the technology development, challenges and translational impact of preclinical PET and SPECT imaging in small animal. It is an interesting topic and clear organization through the manuscript. But the review needs a revision, before its acceptance.
Comments:
- Delete the table 1, theses radiopharmaceuticals and their applications are well known, and Table 1 is not relevant to the theme of the review.
- Figure 1 to Figure 3, these figures need to be enlarged, and increase font size in these figures so that the words can be clearly read.
- Some sentences are too long throughout the manuscript. Too long sentence is not recommended, they should be concise. For example, line 87-90, line 129- 132, line 140-43, line 352-354 (no clear), etc.
- Page 9, two 3.3?
Author Response
Comments 1: Delete the table 1, theses radiopharmaceuticals and their applications are well known, and Table 1 is not relevant to the theme of the review.
Response 1: We thank the Reviewer for the comment. Although we acknowledge that the content of Table 1 is basic and well known to experts, we prefer to retain it as a concise summary that may be helpful for readers less familiar with the field. In addition, we have expanded the manuscript with a new paragraph on theranostic radionuclides in response to another Reviewer’s request, which is closely related to the content of Table 1 and further supports its role as an introductory and guiding element in the review. We believe that retaining Table 1 improves the clarity and accessibility of the review without detracting from its scientific depth.
Comments 2: Figure 1 to Figure 3, these figures need to be enlarged, and increase font size in these figures so that the words can be clearly read.
Response 2: We thank the Reviewer for this helpful suggestion. Figures 1–3 have been revised by enlarging their overall size and increasing the font size of all labels and annotations to improve readability. The updated figures are now clearer and easier to read both on screen and in print.
Comments 3: Some sentences are too long throughout the manuscript. Too long sentence is not recommended, they should be concise. For example, line 87-90, line 129- 132, line 140-43, line 352-354 (no clear), etc.
Response 3: We thank the Reviewer for pointing out this issue. In response, the manuscript has been carefully revised to shorten overly long sentences and improve clarity throughout the text. The specific sections have been rephrased into shorter, more concise sentences, and additional stylistic edits were applied across the manuscript to enhance overall readability. All the changes are highlighted in the main text (in blue).
Comments 4: Page 9, two 3.3?
Response 4: We appreciate the Reviewer for noticing this formatting issue. The duplicated section numbering (“3.3”) on page 9 has been corrected, and the section headings have been renumbered accordingly in the revised manuscript.
Reviewer 2 Report
Comments and Suggestions for Authors
The reviews are always have a rol to represent the state of art of the selected scientific topic. This review is filling well that rol, however the historical overview a little bit too much detailed, but never mind! So I am really satisfied wih the manuscript, only a few comment i want to address to the athors, before agreeing with the publication by the following:
1, Please explain a little bit more wide list of the radioisotopes and put the light also for the theranostic aspects. I mean such pair of isotopes as 44Sc/47Sc, 152,155Tb/161Tb, 203Pb/212Pb, 124I/211At, 206Bi/213Bi.
2, Please tell some sentence about the practical applications of zebrafish in PET imaging. According to the resolution limit of severeal PET isotopes heavy to image how can detect any valuable informations on that small zebrafish.
3, I am sure the authors can find already a publications referring about the artificial intelligence in this field. Please add your opinions about the rol of the AI in the future.
Author Response
We thank the Reviewer for the valuable and constructive comments, which have helped to improve the quality and clarity of the manuscript. We have carefully addressed all remarks and revised the manuscript accordingly. The specific responses are detailed below.
Comments 1: Please explain a little bit more wide list of the radioisotopes and put the light also for the theranostic aspects. I mean such pair of isotopes as 44Sc/47Sc, 152,155Tb/161Tb, 203Pb/212Pb, 124I/211At, 206Bi/213Bi.
Response 1: We thank the Reviewer for this important suggestion. In response, we have expanded the manuscript by adding a dedicated paragraph (2.3. Emerging Theranostic Radionuclide Platforms for Preclinical Imaging and Therapy) that discusses element-matched theranostic radionuclide pairs in greater detail. The revised text now includes an overview of the ⁴⁴Sc/⁴⁷Sc, ¹⁵²,¹⁵⁵Tb/¹⁶¹Tb, ²⁰³Pb/²¹²Pb, ¹²⁴I/²¹¹At, and ²⁰⁶Bi/²¹³Bi pairs, highlighting their diagnostic and therapeutic characteristics, chemical compatibility, and translational potential in preclinical nuclear medicine.
Comments 2: Please tell some sentence about the practical applications of zebrafish in PET imaging. According to the resolution limit of severeal PET isotopes heavy to image how can detect any valuable informations on that small zebrafish.
Response 2: We thank the Reviewer for this comment. In response, we have added a paragraph describing the practical applications of zebrafish (Danio rerio) in preclinical PET imaging. The revised text explains how limitations related to the small size of zebrafish and PET spatial resolution can be mitigated using high-sensitivity PET/CT systems, optimized imaging setups, and dedicated immobilization platforms. We also discuss the use of physical and digital zebrafish phantoms to validate imaging protocols and demonstrate that meaningful quantitative and anatomically correlated PET data can be obtained despite resolution constraints.
Comments 3: I am sure the authors can find already a publications referring about the artificial intelligence in this field. Please add your opinions about the rol of the AI in the future.
Response 3: We thank the Reviewer for raising the important point regarding the potential role of artificial intelligence (AI) in preclinical PET and SPECT imaging. While AI-based methods are developing very rapidly and their exact future role cannot be reliably predicted at present, we believe that such approaches are likely to find broad applications in areas such as image reconstruction, quantitative analysis, multimodal data integration, and quality control. However, a major current limitation is the lack of unified guidelines, regulatory frameworks, and institutional recommendations defining the implementation and validation of AI-based methods in PET and SPECT imaging. We have therefore added a brief, forward-looking statement to the manuscript (5.4. Role of Artificial Intelligence in Preclinical PET and SPECT Imaging) that acknowledges both the potential of AI and the current absence of standardized regulations governing its use in preclinical molecular imaging.
We believe that these revisions have significantly strengthened the manuscript and addressed all concerns raised by the Reviewer. We are grateful for the constructive feedback and hope that the revised version meets the Reviewer’s expectations.